# Eco-Friendly Dispersant-Free Purification Method of Boron Nitride Nanotubes through Controlling Surface Tension and Steric Repulsion with Solvents

**DOI:** 10.3390/nano13182593

**Published:** 2023-09-19

**Authors:** Minsung Kang, Jungmo Kim, Hongjin Lim, Jaehyoung Ko, Hong-Sik Kim, Yongho Joo, Se Youn Moon, Se Gyu Jang, Eunji Lee, Seokhoon Ahn

**Affiliations:** 1Institute of Advanced Composite Materials, Korea Institute of Science and Technology (KIST), Jeonbuk 55324, Republic of Korea; t19132@kist.re.kr (M.K.);; 2School of Materials Science and Engineering, Gwangju Institute of Science and Technology (GIST), Gwangju 61005, Republic of Korea; 3Nano Hybrid Technology Research Center, Korea Electrotechnology Research Institute (KERI), Changwon-si 51543, Republic of Korea; 4Department of Quantum System Engineering, Jeonbuk National University, Jeonju-si 54896, Republic of Korea

**Keywords:** BNNT, purification, eco-friendly, cosolvent, interfacial energy

## Abstract

Boron nitride nanotubes (BNNTs) were purified without the use of a dispersant by controlling the surface tension and steric repulsion of solvent molecules. This method effectively enhanced the difference in solubilities of impurities and BNNTs. The purification process involved optimizing the alkyl-chains of alcohol solvents and adjusting the concentration of alcohol solvent in water to regulate surface tension and steric repulsion. Among the solvents tested, a 70 wt% t-butylalcohol in water mixture exhibited the highest selective isolation of BNNTs from impurities based on differences in solubilities. This favorable outcome was attributed to the surface tension matching with BNNTs, steric repulsion from bulky alkyl chain structures, and differences in interfacial energy between BNNT–liquid and impurity–liquid interfaces. Through this optimized purification process, impurities were removed to an extent of up to 93.3%. Additionally, the purified BNNTs exhibited a distinct liquid crystal phase, which was not observed in the unpurified BNNTs.

## 1. Introduction

The boron nitride nanotubes (BNNTs), which were first synthesized by Zettl’s research team in 1995, have attracted the attention of many researchers, owing to their tubular structure analogous to that of carbon nanotubes (CNTs) [1]. Both BNNTs and CNTs exhibit high temperature stability and chemical resistance due to their similarity in crystal structure (i.e., hexagonal structure). On the other hand, BNNTs offer distinct advantages over CNTs, such as neutron shielding, thermal neutron capture, and piezoelectric properties, as they are composed of boron and nitrogen [2,3,4,5]. For this reason, BNNTs can be used for applications in aerospace composite, energy harvesting materials, and boron neutron capture therapy [5,6,7,8]. In order to facilitate the utilization in such fields, a mass production strategy for the BNNTs was developed by the National Research Council Canada (NRC) research team. Their pioneering works demonstrated the successful fabrication of the few-walled BNNTs with high crystallinity using the high-temperature RF-plasma methods [9]. However, the presence of BN-related impurities, such as hexagonal boron nitride (hBN), BN-cage, and amorphous BN, in the produced BNNTs limits further applications, requiring additional purification processes [9,10].

To solve this problem, many methods for purifying BNNTs are being researched and developed. The research team from NRC developed a method for purifying BNNTs using chlorine gas. In the process, the BNNTs are exposed to chlorine gas at high temperatures (i.e., 950 °C) in a tube furnace, which can remove impurities and obtain the purified BNNTs. However, it requires careful handling and can be environmentally harmful because of the high toxicity of the chlorine gas [11]. As another purification method, Professor Pasquali’s research team suggested the possibility of using chlorosulfonic acid (CSA) as a dispersible solution and purification for BNNTs without additional dispersants. Through this purification process, purified BNNTs can maintain their intrinsic properties due to the absence of residual impurities on their surface. However, handling CSA requires a lot of attention to safety since it is an extremely strong acid, which can cause serious corrosion or explosive reactions with certain reactants [12,13]. Alternatively, using dispersants such as polymer, surfactant, and biomolecules to disperse and purify BNNTs is relatively less hazardous and easier compared to the aforementioned methods. This use of dispersants has the greatest advantage of being able to easily obtain the high-purity level of the BNNTs. However, organic residues on refined BNNT surfaces remain a problem because this approach can adversely affect the performance of various applications [14,15,16,17]. Although these residues can be removed by the thermal decomposition process, the process is economically and environmentally unfavorable [17,18]. To avoid the aforementioned issues, various attempts have been made to purify BNNTs using only general solvents, but only heptane has shown successful results [19]. This indicates that research on BNNTs purification methods using organic solvents is currently inadequate.

In this study, we aimed to develop a simple and effective dispersant-free purification process using solvent engineering via the optimization of cosolvent configuration. We utilized two concepts: (i) the formation of a stable dispersion of BNNT and (ii) the effective separation of BNNTs and hBN impurity using the disparity of dispersion efficiency. The thermodynamically stable nanotube dispersion requires a negative mixing Gibbs free energy between the solvent and nanotubes (ΔG_mix_), which can be achieved by minimizing the difference between the nanotube and solvent surface energies and ensuring a small or negative mixing enthalpy of the solvent and nanotubes (ΔH_mix_) [20,21,22,23]. Therefore, our primary objective was to identify a solvent with the appropriate surface tension for effective BNNTs dispersion. Additionally, during the dispersion of nanomaterials in a solvent, the steric repulsion, influenced by the size of the solvent molecules, might aid in overcoming the Leonard-Jones potential and improving the stability of the dispersion [23,24,25,26]. Then, the difference in interfacial energy between BNNTs and hBN impurities originating from structural differences is expected to exhibit distinct dispersion characteristics. Thus, we hypothesized that if the surface tension of the solvent is optimized for BNNTs dispersion, the steric repulsion of solvent molecules might maximize the difference in the solubility of BNNTs and hBN-like impurities as shown in Figure 1. It has been reported that alcohol molecules are effective solvents for BNNT dispersion [23] and that surface tension can be controlled by the cosolvent concentration and alkyl chain structure of alcohol molecules [27]. Additionally, the size of the molecular weight and steric repulsion, determined by the alkyl chain structure, can influence dispersion. Moreover, it has been reported that a mixture of alcohol and distilled water can function similarly to liquid surfactants, in contrast to pure solvents [28,29]. Based on this knowledge, the dispersion of BNNTs in alcohols along with varying concentrations of alcohol in water were investigated. As a result, it was found that alcohol molecules with high molecular weight and bulky alkyl structures at 70 wt% concentration exhibited a better match in surface tension with BNNTs. Additionally, it was observed that the dispersion was maximized due to the difference in interfacial energy between BNNTs and hBN impurities. We utilized the explored mechanism to conduct the purification of BNNTs, achieving the removal of hBN impurities up to 93.3% without the use of any dispersants. Furthermore, the purified BNNTs displayed a well-defined liquid crystal phase, which was not observed in the unpurified BNNTs.

## 2. Materials and Methods

### 2.1. Materials

The boron nitride nanotubes (BNNTs) were prepared via the high-temperature RF-plasma methods using h-BN with a particle size of 70 nm as the BN source precursor at the High-enthalpy Plasma Research Center at Jeonbuk National University (Wanju, Republic of Korea). The tert-butanol (t-BA) was purchased from Junsei. Methanol, ethanol, 1-propanol, and iso-propanol were purchased from Daejung chemicals. The commercial hexagonal boron nitride (hBN) samples in sizes of 70 nm, 200~500 nm, and over 1 μm were purchased from MKnano, Momentives, and Denka, respectively. The impurities were collected by recovering and drying the precipitated materials after purification. The sodium cholate hydrate from Sigma-Aldrich was used for liquid crystal experiments.

### 2.2. Purification Processes

The as-synthesized BNNTs contain a significant amount of amorphous boron, which can be easily removed by thermal treatment and a hot water washing process. The pristine materials were introduced to thermal oxidation treatments at 670 °C in the air for at least 6 h [30]. After the calcination process, this material changed from gray to snow-white. This means that amorphous boron (a-Boron) has been successfully converted to B_2_O_3_. Since B_2_O_3_ is easily soluble in methanol or deionized water (DIW), the samples were washed with methanol or DIW. The a-Boron-removed BNNTs samples were dried by freeze-drying. The dried samples were dispersed in t-BA (70 wt%)/DIW solution at a concentration of 0.25 mg/mL. The BNNTs dispersion was left to settle for 24 h to remove large impurities, and then the upper solution was centrifuged at 10,000 rpm for 60 min (CF-10, Daihan Scientific, Wonju, Republic of Korea). Following centrifugation, the supernatant containing purified BNNTs was concentrated using a rotary evaporator, and purified BNNTs were obtained through freeze-drying.

### 2.3. Dispersion Test

The solvent mixtures were prepared by varying the concentration of alcohol. All dispersions of BNNTs, impurities, and commercial hBNs were prepared at a concentration of 0.2 mg/mL through bath sonication for 30 min. Subsequently, these solutions underwent centrifugation at 4100 rpm for 30 min, and only the supernatant was collected for a measurement of absorbance using a UV–Vis–NIR spectrometer. Absorbance was measured in the 380–500 nm range (Appendix A). Neither BNNTs nor hBNs exhibited any specific absorption peaks in this range, thus only the values measured at 400 nm were selected. Additionally, the pure t-BA sample was excluded from the measurement due to its low melting point (25 °C), as it could easily freeze during the centrifugation process.

### 2.4. Liquid Crystal

The BNNTs paste for BNNTs liquid crystal experiments was prepared by referring to the previously reported paper [31]. Sodium cholate hydrate was used as a dispersant to prepare aqueous dispersions of both unpurified and purified BNNTs. The dispersions were concentrated using a rotary evaporator and bath-sonicator. The concentration of the final BNNTs paste was approximately 1.8 wt%. The liquid crystal phase was observed using a polarized optical microscope equipped with a rotary stage (POM, ECLIPSE LV100N POL, Nikon, Tokyo, Japan) at room temperature.

### 2.5. Measurements

The morphology of BNNTs and impurity was investigated using scanning electron microscopy (NanoSEM-460, FEI, Hillsboro, OR, USA), transmission electron microscopy (TEM) (Technai G2 F20, FEI, Hillsboro, OR, USA) at 200 kV and high-resolution TEM (Titan G2 Cube 60-300, FEI, Hillsboro, OR, USA) at 80 kV. The SEM images of the samples were prepared by filtering the solution. The TEM image of the samples was fabricated by dropping the solution on the lacey carbon film on the copper grid. To measure the X-ray diffraction (XRD) of the samples, all XRD samples were prepared by filtering the solution on the PVDF membrane. The prepared samples were analyzed via SmartLab X-ray diffractometer (Rigaku, Tokyo, Japan) using Cu-Kα source (10° ≤ 2θ ≤ 90°). The dispersion ability of solutions was conducted by using a UV–Vis–NIR spectrometer (V670, Jasco, Seoul, Republic of Korea). The thermal gravimetric analysis (TGA) was conducted to confirm the presence of organic substances on the surface of purified BNNTs. The TGA was performed using the Q-50 model (TA instrument), with a sample mass of 5–10 mg and a heating rate of 10 °C/min under a nitrogen atmosphere.

## 3. Results and Discussion

### 3.1. Optimized Purification Process to Control the Surface Tension and Steric Repulsion of Solvents

To control the surface energy of cosolvent, the concentration in water and alkyl structures of alcohol molecules were varied to find the optimized BNNTs dispersion conditions. Figure 1 shows the difference in dispersion efficiency between the BNNTs and hBN impurities depending on the choice of alcohol. The BNNTs and impurities were dispersed in various alcohol/DIW mixtures (i.e., methanol, ethanol, 1-propanol, iso-propanol (IPA), tert-butanol (t-BA)), and the differences in the absorbance at 400 nm using UV–NIR spectroscopy was observed. In the case of methanol, ethanol, and 1-propanol, there is no significant difference in the absorbance between BNNTs and impurities. However, when using IPA and t-BA as cosolvent, the difference in absorbance between BNNTs and impurities is significantly increased. Figure 1f demonstrates how the solvent’s steric structure and molecular weight (MW) affect the dispersion of BNNTs. For linear alcohol molecules, the increase in maximum absorbance depending on the MW is negligible. However, when comparing 1-propanol with IPA, which has a branched structure, the significant steric repulsion effect of the solvent size effect on BNNTs dispersion is observed [24,29].

This effect was further maximized in t-BA, which has an additional methyl group compared to IPA. Among the solvents tested, t-BA exhibited the most pronounced steric repulsion. Consequently, t-BA/DIW was selected as the solvent mixture with the highest potential for dispersing and purifying BNNTs in our study. Based on these results, the dispersibility test of BNNTs in different concentrations of t-BA/DIW mixtures was conducted (Figure 2). One hour after bath sonication, a significant amount of BNNTs were aggregated when the t-BA ratio was below 40 wt%, while the BNNTs remained well dispersed in the range of 60–90 wt% t-BA/DIW solutions. To determine the optimal condition for BNNTs purification, the dispersions were left to settle for 7 days (Figure 2a). After 7 days, it was found that excellent dispersion stability was achieved in the t-BA (70 wt%)/DIW solution, whereas BNNTs aggregation occurred in other concentrations. These results indicate the possibility of producing a stable BNNT dispersion using the in t-BA (70 wt%)/DIW solution as the optimal condition for purification. SEM analysis was conducted to examine the morphology differences between the supernatant and aggregates of the BNNTs solution in 40 wt%, 70 wt%, and 90 wt% t-BA/DIW solution, respectively (Figure 2b). The supernatant solution displayed BNNTs and small-sized impurities, with no significant difference observed among different t-BA/DIW solutions. However, the sediments exhibited distinct differences. In the SEM image of the sediments from the 40 wt% and 90 wt% t-BA/DIW solution, a high content of BNNTs with the impurities was observed due to the low dispersion of BNNTs under these conditions. In contrast, in the sediments of the t-BA (70 wt%)/DIW solution, relatively large aggregates of hBN impurities with a size range of up to 1 µm were predominantly observed. These results, deduced from the observations in Figure 1e and Figure 2a, can be attributed to the difference in dispersion forces between impurities and BNNTs. In the 70 wt% tBA/DIW mixture, BNNTs form a highly stable dispersed phase due to the molecular weight effect and steric repulsion effect. Conversely, impurities struggle to form a relatively uniform dispersed phase, leading to their easy precipitation, as evident in Figure 2b. This suggests that it is possible to primarily separate BNNTs from impurities (~1 µm) in t-BA (70 wt%)/DIW solution. However, there were still numerous impurities with sizes smaller than 400 nm remaining.

To remove these impurities, the supernatant was centrifuged at 10,000 rpm for 1 h. The resulting supernatant (Figure 3a) showed the predominant presence of BNNTs, with only a minimal presence of impurities smaller than 100 nm. In contrast, the sediments (Figure 3b) contained both impurities and BNNTs. To understand the mechanism for the effectiveness of the respective purification processes in removing impurities, the size of the impurity samples obtained by two different methods was analyzed in detail (Figure 3c). The primary impurity (first impurity) is sediment obtained by allowing the dispersed BNNTs solution to settle for one day. The secondary impurity (second impurity) is the sediment that settled after the centrifugation of the BNNT solution (after the primary removal) at 10,000 rpm. In the first impurity, the impurity particles of various sizes were observed with the largest proportions of hBN impurities larger than 1 µm. In the second impurity, impurities smaller than 0.2 µm were dominant. These results suggest that the size of the impurities can be related to the difference in the removal efficiencies, which is likely to show different interfacial interactions with the dispersing solvent molecules.

In order to clarify the relationship of BNNTs and different sizes of hBNs (~70 nm, 200~500 nm, over 1 µm) with the surface energy of the dispersing solvents, the dispersion state (represented by absorbance) was plotted against the t-BA/DIW mixtures. It was observed that BNNTs exhibited good dispersion at a surface tension range of 20.9–22.0 mN m^−2^. The hBN particle sizes below 500 nm were also well dispersed within a surface tension range of 20.4–23.2 mN m^−2^, while the particles larger than 1 μm showed poor dispersion (Figure 3d). Furthermore, at certain concentrations of alcoholic solvents and water, alcoholic solvents can act similarly to surfactants, helping to disperse nanomaterials. At these specific concentrations, these solvent molecules can also interact with water to form dimers or trimers, which can help overcome the van der Waals forces between nanomaterials [24,26,28,29,32]. This difference in dispersion is predicted to be attributed to the structural differences between BNNTs and hBN. The hBN structure is similar to graphite with a [002] plane structure and a well-ordered 3D structure due to electrostatic interactions between each layer. On the other hand, BNNTs have a one-dimensional structure in the form of a nanotube; thus, they have weaker interactions than hBN. For this reason, hBN is subject to a relatively smaller solvent effect than BNNTs. As a result, the dispersion characteristics of BNNTs can be relatively better than those of hBN. Furthermore, hBN exhibited different dispersion characteristics depending on the size of the particles. Smaller particles were found to have a higher tendency to disperse than larger particles. Based on these results, we confirmed the possibility of BNNTs purification using a specific mixing ratio of cosolvent. By leveraging the surface tension of the solvent and steric repulsion caused by the size of the solvent molecules, and taking into account the disparities in dispersion stability between BNNTs and hBN impurities, our newly developed dispersant-free purification method demonstrated the capability to remove a substantial portion of hBN impurities. Then, TGA analysis was carried out to examine the presence of organic impurities on the surface of the purified BNNTs (Appendix A). BNNTs, both before and after purification, were measured in nitrogen at a heating rate of 10 °C/min to 900 °C. The weight of both BNNTs samples did not show any difference, indicating the absence of organic materials on the BNNTs surface after purification.

### 3.2. Characterization and Comparison of Purified BNNTs

Figure 4 explains the difference in XRD patterns between the unpurified BNNTs (B_2_O_3_ removed BNNTs) and purified BNNTs (at 5000 rpm and 10,000 rpm). In the unpurified BNNTs, two peaks are shown; a relatively broad and weak peak corresponding to the BNNTs (2θ = 25.8°); and a sharp and strong peak corresponding to the hBN (2θ = 26.8°). The sharpness and the high intensity of the hBN peak originate from its well-defined stacked structure. On the other hand, BNNT, as a one-dimensional (1D) material, features a BN structure akin to hBN. Moreover, it typically consists of very few walls, averaging between 2 and 10 walls. SEM analysis (Figure 3a) has further revealed that BNNTs are arranged in a random manner. Consequently, due to their relatively low wall count and randomly structured arrangement, the peaks associated with BNNTs are observed to be broader and weaker in comparison to those of hBN. [19,33]. This phenomenon is consistent with the XRD results of BNNTs purified using various purification methods [19,31,34,35]. As the purification progressed, it was observed that the intensity of the BNNTs-derived peak increased while the hBN-derived peak decreased. In order to confirm the relative purification efficiency of this system, calculations based on the previously reported equation (shown in Appendix A were performed [31]. Appendix A shows the deconvoluted patterns and calculated values obtained from the Gaussian fitting of the XRD patterns of each sample (unpurified and purified BNNTs (at 5000 rpm and 10,000 rpm)). It was calculated that the purification percentile of the BNNTs rises to 55.9% at 5000 rpm and to 93.3% at 10,000 rpm. Nevertheless, the purified BNNTs at 10,000 rpm exhibit a small shoulder peak at 26.7°, which might be caused by the nano-sized hBN particle and BN-cage [35].

In order to analyze the origin of the small shoulder peak at 26.8°, TEM analysis was conducted on the BNNTs samples (Figure 5). The most remained impurities exhibited a shell-like structure (BN-cage). In particular, many impurities showed an open structure on one side (Figure 5b), while others were observed to be connected to the wall of the BNNTs (red dashed lines in Figure 5c). The morphology of these cage impurities is associated with the synthesis mechanism of BNNTs. The synthesis mechanism of BNNTs has been reported as the root growth process, where liquid boron droplets are formed at the beginning of the reaction. The B_x_N_y_H_z_ species rapidly adsorb and grow on this droplet, leading to the synthesis of nanoparticle-linked BNNTs [10,36]. Since the boron in the nanoparticles can be easily removed via heat treatment and washing with DIW or MeOH, only hollow structures are typically observed at the end of BNNT growth. Because these nano-sized BN cages formed during the BNNT synthesis process have a similar dispersion with BNNTs and are covalently connected to BNNT in some cases, BN-cage impurities could not be separated from BNNTs through the mild solvent purification method used in the present study.

In spite of the existence of small portion BN-cage impurities, lyotropic liquid crystals could be obtained by using purified BNNTs over 90% purity, whereas unpurified BNNTs could not form liquid crystals because the alignment of BNNTs is disturbed by impurities.

Figure 6a shows the POM image of unpurified BNNTs. The unpurified paste had low birefringence, an indication that the alignment of the BNNTs had been disturbed by impurities. In contrast, liquid crystal was observed in purified BNNTs, showing that the purified BNNT was aligned in a specific direction in an appropriate solvent (Figure 6b). This phenomenon is related to the mobility of nanotubes at a specific concentration. Above a critical concentration, the concentrated isotropic phase becomes unstable, restricting the movement and rotation of nanotubes and leading to the formation of spontaneously aligned liquid crystal phases [37]. This critical concentration is correlated with the purity of BNNTs [30,31,38]. High-purity BNNTs can exhibit liquid crystal phases at lower concentrations. For BNNTs with exceptionally high purity and crystallinity, the liquid crystal phase has been observed at concentrations exceeding 0.8 wt% [38]. From this perspective, the newly developed purification method enables the purification of BNNTs with high purity, allowing them to form liquid crystal phases at low concentrations, despite the presence of small-sized BN cages.

Finally, several previously reported purification methods are summarized in Table 1. Compared to existing purification methods, our new purification methods are believed to be highly beneficial for researching the medical or intrinsic properties of BNNTs because they enable purification without the use of dispersants.

## 4. Conclusions

We developed a surfactant-free and eco-friendly purification process for BNNTs by simply using a mixture of common alcohol and water solvents. The main focus of this purification process was to control the surface tension of solvents and utilize the steric repulsion effect. By utilizing a t-BA/DIW mixture, we observed differential dispersibility between BNNTs and impurities, attributed to the interplay of surface tension and steric repulsion. Building upon this dispersibility difference, we successfully extracted purified BNNTs from the initially synthesized mixture containing both BNNTs and BN-derived impurities. Although nano-sized BN-cage impurities could not be perfectly removed, purified BNNTs showed the successful formation of liquid crystals, supporting the high purity level. This surfactant-free and eco-friendly purification process would provide significant insights for the advancement of BNNT applications.

## Data Availability

Data underlying the results presented in this paper are not publicly available at this time but can be obtained from the authors upon reasonable requests.

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
