# Peer review of "Eco-Friendly Dispersant-Free Purification Method of Boron Nitride Nanotubes through Controlling Surface Tension and Steric Repulsion with Solvents"

_nanomaterials, 2023, doi:10.3390/nano13182593_

Round 1

Reviewer 1 Report

Boron nitride is a substance and material that has great prospects in various fields of science and technology. A number of factors related to the properties of this material prevent widespread implementation. The authors' work focuses on the problem of purification of boron nitride nanotubes from impurities, which are the same substance, but represented in other morphologies, such as hexagonal boron nitride, BN-cage, and amorphous BN. The authors presented an original method for cleaning, solve the task and also use a green approach, which is deprived of previously used methods available in the literature. The manuscript will be ready for Nanomaterials after correcting the shortcomings indicated below in the comments.

1.     Line 21. Abbreviations without decoding are hardly appropriate in the abstract.

2.     Section 2.3. No explanation as to why a wavelength of 400 nm was chosen for the Absorption measurement? There is also no such data in ECM, as well as the UV-vis absorption spectrum, which is also desirable to give.

3.     It is better to divide the large and main section of the Results and Discussion into logical subsections according to the conducted research.

4.     Figure 1. (a-e) the dimension on the X-axis is not specified. The resulting graph (f): there is no connecting line between the points corresponding to 1-propanol and 2-propanol, why? This is the most demonstrative transition, confirming the huge contribution of steric repulsion effect at the same MW value. It was also good to point out that the absorption measurements were carried out at a wavelength of 400 nm, it is better to add this for all figures where such measurements are made.

5.     Figure 2. The names of the samples are vague and a font is quite small.

6.     Figure 3. (c) all inscriptions are extremely unclear and not visible at all. Both graphs on figure (d), and this is an important and interesting result, should be moved to a separate figure and make it larger. Figure (e), TGA analysis, since the results are superimposed on each other, can be moved to ESM.

7.     Figure 4. and Lines 261-269. Apparently, the impurities are crystalline samples, and on the contrary, BNNTs are, according to the blurred peak, amorphous. If this is the case, then it is good to give the values значения the sizes of the crystallites for BNNTs sample.

8.     Figure 5. Names on the dimension scale are fuzzy and blurred on all figures. You need to increase the font.

9.     Figure 6. It is usually recommended to indicate the temperature at which the formation of the liquid crystal phase was observed.

10.  It is recommended that the authors provide a final comparison table at the end of the Results and Discussion section of the investigated purification method with those studied earlier in the literature, according to the Introduction section. Here, in optional (this is at the discretion of the authors), green metrics for the synthesis and purification process (atom economy of the synthesis, E-factor, yields, use of green solvents, toxicity,…) can also be given. This can make a future article even more interesting for readers.

11 It is recommended to correct the list of references: the full names of the authors are given in some references, incorrect abbreviations are found in more than half of the references.

In general, a good study has been done, the text reads mostly well.

Author Response

Step-by-step response to referees

We thank the reviewers for their thoughtful comments and suggestions on our manuscript. We carefully revised the manuscript. All issues raised by reviewers have been carefully addressed, as attached below. Revisions and supplementary information in the manuscript are highlighted. We hope that these modifications will make the study more suitable for publication in Nanomaterials. Response to Comments We thank the reviewers for their thoughtful comments and suggestions on our manuscript. All issues raised by reviewers have been carefully addressed, as attached below.

Reviewer 2 Report

Authors have examined the preparation of boron nitride nanotube dispersions by different mixed solvents ; however, the discussion is simplified, making it difficult to comprehend the underlying reasons behind the process. It would be beneficial if they could offer a more detailed explanation regarding the correlation between the mixed solvent ability and the dispersion of the nanotubes, using either descriptive expression or visual diagrams . Additionally, in Figure 2b, there appears to be a significant disparity in the sediment's appearance of 75% sample compared to the others, what is the reason.

Author Response

(The authors gave the same response as above.)

Round 2

Reviewer 1 Report

The authors presented a revised manuscript and responses to the reviewer's comments. The form and themselves the responses to the reviewer's comments exceed all expectations. It is with great satisfaction that I recommend the paper to be accepted to Nanomaterials and published as soon as possible.